# Tissue Engineering in Gynecology

**DOI:** 10.3390/ijms232012319

**Published:** 2022-10-14

**Authors:** David Brownell, Stéphane Chabaud, Stéphane Bolduc

**Affiliations:** 1Centre de Recherche en Organogéneèse Expérimentale/LOEX, Regenerative Medicine Division, CHU de Québec-Université Laval Research Center, Québec, QC G1J 1Z4, Canada; 2Division of Urology, Department of Surgery, CHU de Québec-Université Laval, Québec, QC G1V 4G2, Canada; 3Department of Surgery, Faculty of Medicine, Laval University, Québec, QC G1V 0A6, Canada

**Keywords:** tissue engineering, ovary, uterus, vagina, gynecology

## Abstract

Female gynecological organ dysfunction can cause infertility and psychological distress, decreasing the quality of life of affected women. Incidence is constantly increasing due to growing rates of cancer and increase of childbearing age in the developed world. Current treatments are often unable to restore organ function, and occasionally are the cause of female infertility. Alternative treatment options are currently being developed in order to face the inadequacy of current practices. In this review, pathologies and current treatments of gynecological organs (ovaries, uterus, and vagina) are described. State-of-the-art of tissue engineering alternatives to common practices are evaluated with a focus on in vivo models. Tissue engineering is an ever-expanding field, integrating various domains of modern science to create sophisticated tissue substitutes in the hope of repairing or replacing dysfunctional organs using autologous cells. Its application to gynecology has the potential of restoring female fertility and sexual wellbeing.

## 1. Introduction

The female gynecological tract is responsible for the creation of new life; from oocyte production, intercourse, fertilization, embryo implantation and maturation, up until delivery of a fully formed being. Various defects of gynecological structures, either congenital or acquired, can be cause for female infertility. Defined as the inability to conceive after 12 months of frequent coitus, female infertility affects approximately 13% of women [1]. This number increases significantly after the age of 35, with one study reporting infertility rates of 32- and 38-year-old women as 12% and 20%, respectively [2]. With modern lifestyles prioritizing the education and career development of young women, developed parts of the world see trends of women delaying childbearing to later in life, increasing the demand for assisted reproduction technologies (ART). The most common reasons for loss of reproductive organ function are reproductive organ neoplasia or their current treatments, including, chemotherapy, radiotherapy, and surgical resection or total removal of reproductive structures. Adhesions, fibrosis, and age-related complications are also highly prevalent [3]. Congenital malformations should not be overlooked when evaluating potential gynecological solutions. Aside from obvious infertility complications, psychological implications should be taken into consideration, as women with gynecological complications often suffer distress related to female identity and inability to be a “natural” mother despite current ART such as in vitro fertilization (IVF) and surrogacy.

Novel solutions to gynecological organ dysfunction, such as tissue engineering, are increasing in prevalence. Such solutions have the potential to restore organ function by partial or complete organ replacement, sometimes in conjunction with current ART. Tissue engineering solutions mostly rely on the use of autologous cells to recreate tissues in vitro with native-like histoarchitecture that can be grafted to a patient to repair or fully replace a defective organ, often employing a natural or synthetic scaffold to aid in structural support of these constructs. Autologous constructs would allow for genetic motherhood, as well as limited risk of rejection, both by the host or of the embryo during potential implantation/gestation in the uterus. Restoration of a woman’s organ function has the advantage of avoiding ethical dilemmas related to use of surrogate mothers, as well as improving a patient’s sense of womanhood by becoming a “natural” mother. Other than fertility preservation, gynecological organ function restoration improves quality of life. Women having undergone experimental uterine transplant have a restored menstrual cycle, and women having undergone treatment for vaginal aplasia are capable of vaginal intercourse, improving a woman’s psychosexual wellbeing [4]. 

In this review, we present state-of-the-art tissue engineering solutions as treatments for pathologies of the ovary, uterus, and vagina. Only tissue engineering models tried in vivo are evaluated to give an overview of common practices and real potential as clinical solutions. Organ anatomy and common pathologies will be described, as well as current lines of treatment and their pros and cons. Most promising alternatives to current treatment methods will be discussed and compared to the advantages presented by tissue engineering. 

## 2. Tissue Engineering in Gynecology

### 2.1. Biomaterials

Tissue engineering relies mainly on implantation of scaffolds with or without cells. Except for the self-assembly technique or the use of decellularized tissues, creation of scaffolds to engineer female genital tract tissues is currently based on the use of various biomaterials. Some of them are synthetic, others are natural. From in vivo studies to date synthetic biomaterials polyethylene glycol-vinyl sulfonate (PEG-VS), poly(glycolic acid) (PGA), and poly(lactic co-glycolic acid) (PLGA) have been used. Natural biomaterials: alginate, collagen, gelatin, fibrin, and hyaluronic acid have been used alone or in various combinations. These molecules are polymers represented in Figure 1. All of these molecules are authorized by the United States Food and Drug Administration (FDA).

The main advantage of synthetic biomaterials is their low cost and the highly tunable nature of their macro- and micro-structures in terms of form, porosity, mechanical properties, biodegradability, and ability to bind to various molecules of biological interest. First produced in 1859, polyethylene-glycol (PEG) is a hydrophilic polymer used for diverse applications. It is a low molecular weight, linear polyether polymer consisting of ethylene glycol monomers. Polyethylene-glycol hydrogels are considered as inert and safe. Their large-scale use by the pharmaceutical industry makes these molecules a product of choice for tissue engineering, especially due to their capacity for surface modification, bioconjugation, and drug delivery [5]. Their branched forms can be combined with vinyl-sulfone (VS) groups. Vinyl sulfone is an organic electrophilic component reactive to thiol residues. The first synthesis of PEG-VS was published in 1996 [6]. Polyglycolic acid is an aliphatic polyester polymer, hydrophobic with a high rate of biodegradation. This characteristic does not allow for supporting cell growth or enough extracellular matrix (ECM) deposition to maintain adequate mechanical properties. For this reason, PGA is often use with copolymers. Polyglycolic acid degradation releases lactic acid that can be detrimental to the cell microenvironment through extracellular matric (ECM) remodeling and cell signaling [7]. Polylactic co-glycolic acid is a copolymer of poly lactic acid (PLA) and PGA. The degradation rate of PLGA is tunable but longer than PGA alone. The more lactic acid comprised in the polymer, the slower it degrades [8]. 

Alginate is a natural polymer derived from seaweed, so it is not found in the animal ECM. It is a polysaccharide composed of monomers of mannuronic acid and guluronic acid. Hydrogels are obtained after divalent ions, most often calcium, allow cross-linking of the alginate. The surface of alginate can be funtionalized to serve as a scaffold or the polymer can serve as an encapsulation vessel [9]. 

Contrary to synthetic biomaterials, natural biomaterials found in the animal ECM are more difficult for engineers to use but are often a better choice due to their similarity to ECM molecules, even though they produce less complex scaffolds than native matrixes. Collagens are a family of 28 molecules. Collagen type-I is the main component of ECM and represents 30% of the dry-weight of humans. It is therefore the most obvious candidate for recreating scaffolds for tissue engineering [10]. Collagen can be combined with other biomaterials from synthetic or natural origins to increase the range of its biological functions or its mechanical properties [11,12]. Apart from tissue engineering for the female genital tract, collagen has been used, for example, for the reconstruction of skin [12], bronchi [13,14], or ligaments [15] and allowed the obtaining of tissues histologically and functionally close to native ones. Gelatin is formed by breaking the triple helix structure of collagen into a single-stranded one. The temperature of the gelatin solution easily controls its gelation process. Gelatin is biocompatible and easy to use. It presents some of the advantages of collagen but without having exactly the same properties, especially concerning mechanical properties [16]. Fibrin is the main constituent of blood clots and forms a glue that is widely used as an efficient biological sealant in vivo [17] or to study angiogenesis in vitro in a very simplistic way [18,19]. Fibrinogen is composed of three regions; D region at the extremities; the beta and gamma nodules; and the E region at the center, the central node [20]. Following activation by thrombin, fibrinogen molecules form a biological scaffold on which cells can migrate and proliferate. This structure is also the scene of remodeling events as it occurs during the wound- healing process. Hyaluronic acid is an anionic, nonsulfated glycosaminoglycan found in the ECM, where it contributes to cell proliferation and migration. It is a constituent of the oocyte ECM prior to ovulation. Hyaluronic acid has unique viscoelastic properties as well as good biocompatibility and biodegradability. Hyaluronic acid can be degraded in a natural way by a family of enzymes called hyaluronidases [21]. 

The choice of which biomaterial to use should depend on specific tissue engineering objectives. Scaffolds ideally should have similar mechanical and biochemical properties to their native ECM. This is why collagen is so widely used, as it is omnipresent in the body and often provides good mechanical properties. Caution should be taken when implanting biomaterials in vivo, as foreign-body reactions may be induced by exogenic materials, depending on their surface properties and degradation products, leading to fibrosis [22]. Interestingly, mesenchymal stem cells (MSC) present immunomodulatory properties that have been observed to reduce these foreign-body reactions [23]. However, MSCs alone cannot be used for all tissue-engineering applications [24,25].

### 2.2. Ovaries

#### 2.2.1. Anatomy

The most important organ in terms of female genetic fertility, the ovaries provide an environment for female gamete cells to mature before being expelled towards the fallopian tubes at each menstrual cycle for potential fertilization. The ovaries also act as an endocrine organ responsible for estrogen, testosterone, and progesterone production (Figure 2A). Once colonized by germ cells during gestation, a female will reach a peak of 6–7 million germ cells, after which atrasia initiates and the germ cell number decreases rapidly. At birth, a female will have about one million germ cells, and at puberty she will have only 300,000–400,000. Only 300–400 follicles will ovulate as functional oocytes until menopause, after which only 1000 follicles will be left [26]. The ovaries are composed of a central medulla and an outer cortex enveloped in a columnar epithelium [27]. Follicular cells are found within the ovarian cortex. At each menstrual cycle, follicles mature following endocrine signals, mainly an increase in follicle stimulating hormone (FSH) and luteinizing hormone (LH). As many as 3–11 follicles are fully matured at each cycle, but only 1 oocyte is typically ovulated [28]. The large number of primordial and primary follicles available lend to the possibilities of various reproductive therapies.

#### 2.2.2. Pathologies

Various pathologies can affect the ovaries, influencing both fertility and endocrine function, as well as causing psychological distress [29]. Polycystic ovary syndrome (PCOS) affects around 21.3% of females of reproductive age worldwide [30] and is characterized by irregular menstrual cycles, hyperandrogenism, and polycystic ovaries, representing a significant burden on healthcare systems. In 2020, ovarian cancer was the seventh most common reason for cancer mortality in women globally, responsible for 314,000 new cases and 207,000 deaths [31]. Ovarian cancer often requires aggressive treatment strategies, including ovariectomy. This poses various complications such as the lack of endocrine function of the ovaries as well as absolute infertility if solutions are not available to preserve fertility, such as for juvenile patients with immature ovaries, or in patients requiring urgent intervention. For females undergoing treatment for any cancer, ovarian function is at risk due to the adverse effects of radiotherapy and chemotherapy. These therapies, widely used in combination, present a high risk of ovarian failure at all ages [32]. Women are urged to not delay childbearing once stable in remission following treatment due to the significant long-term effects.

#### 2.2.3. Current Treatments

Current treatment strategies for PCOS rely on hormone balancing, including contraceptives and other hormone-controlling drugs, as well as weight control [33]. Ovulation-inducing drugs are used for PCOS patients when attempting pregnancy. Surgical intervention is rarely advised as most PCOS cases are manageable. On the contrary, sterility due to cancer treatments or ovariectomy is often absolute [32]. The current gold standard in fertility preservation for cancer patients is oocyte harvesting and cryopreservation for later IVF (Figure 2B). Ovulation-inducing drugs are administered in order to harvest multiple oocytes at once. The inconvenience of this method lies in the reliance on postponing cancer treatment while waiting for oocyte retrieval, which usually takes 2–3 weeks [34]. For aggressive cancers, this delay in treatment may significantly increase mortality. There is currently no treatment beyond clinical trials for pre-pubescent female cancer patients. Oocyte retrieval is impossible, as pre-pubescent ovaries are too immature to produce functional oocytes. This means treatment strategies for pre-pubescent girls may be less aggressive in attempts to preserve fertility, therefore potentially increasing mortality. Endocrine function plays an essential role in the development and maintenance of tissue structure [35]. Current treatment relies on estrogen/ progestin replacement therapy to combat the effects of ovarian failure but is related to long-term cardiovascular risks [36]. Effectively, estrogen/progestin replacement fails to recreate the complex endocrine function of the ovaries.

#### 2.2.4. Alternative Solutions

##### Ovarian Tissue Cryopreservation

Currently, several alternative treatments are being developed for ovarian failure [37]. The most prominent alternative solution to date is ovarian tissue cryopreservation. Instead of harvesting mature oocytes, recent advances have put in place protocols for cryopreserving whole tissue slices that can be autografted post treatment (Figure 2C) [32,34,37,38,39]. The tissue may be grafted orthotopically in the pelvic cavity, even within remaining ovaries, allowing for spontaneous pregnancy in 60% of cases [39]. The thawed tissue may also be grafted heterotopically such as subcutaneously. Although the conditions may not be optimal for follicle development, ease of access is an advantage for this procedure when orthotopic sites are not suitable for grafting [40]. Oocytes retrieved from a heterotopic graft in the abdominal wall were successfully fertilized via IVF, leading to the first pregnancy after heterotopic grafting in 2013 [41]. Moreover, ovarian tissue autografting is currently the only proven method of ovarian function preservation for pre-pubescent girls unable to ovulate for oocyte or embryo cryopreservation. The immature follicles will continue to develop once grafted, resulting in mature, viable germ cells. While autografting of frozen-thawed ovarian tissue has proven promising as a one-shot treatment for fertility restoration and endocrine restoration, there remains an intrinsic risk of reintroduction of malignant cells to cancer patients in remission. 

##### Tissue Engineering 

Tissue engineering may provide a solution to eliminate the risk of malignancy (Figure 2D). The first attempt at tissue engineering by Gosden in 1990 [42] cultured murine ovarian cells in an autologous plasma clot, effectively a fibrin scaffold. Sterilized mice were grafted with TE ovaries and fertility was restored, with intercourse often resulting in pregnancy. The grafted mice also showed signs of endocrine recovery. However promising, this first report proved that the murine cells were effectively a mix-bag of various ovarian cells. When aiming to minimize risk of malignancy while using biopsies of cancer patients, there must be diligent sorting of cells and screening for malignant cells. Building from the work of Gosden, various tissue-engineered ovary models have been introduced, employing scaffolds to support follicle maturation in vitro and in vivo murine models [33,43,44]. Plasma clots showed promise in several models [42,45,46], providing a sufficient environment for follicle maturation, and having the advantage of being autologous. In order to improve the manipulability of the models, fibrin scaffolds have been used [47,48,49], the main component of plasma, in order to retain the advantageous properties of fibrin while precisely controlling composition for culture optimization. Indeed, fibrin is an interesting biomaterial for ovarian tissue engineering, as fibrin is deposited in solid tumors and in wound healing prior to angiogenesis and has been demonstrated to promote angiogenesis when implanted in vivo [50]. Angiogenesis is critical in order to maintain follicles imbedded within the engineered ovary. Hydrogel combinations of fibrin with collagen, alginate, or hyaluronic acid (HA) have been used successfully [49,51], retaining the pro-angiogenic properties of fibrin while improving upon mechanical properties of the grafts. Other natural hydrogel scaffolds have also been used, including collagen [52], gelatin [53], and alginate [54,55]; however, alginate scaffolds do not permit graft vascularisation and the follicle population rapidly dies, possibly due to the non-mammalian nature of alginate needing functionalization for mammalian cell culture. The synthetic hydrogel poly(ethylene glycol) vinyl sulfone (PEG-VS) has also been successfully implemented as a scaffold [56]. Interestingly, Laronda et al. implemented 3D printing of a gelatin scaffold to form a microporous engineered ovary [53]. This technique allowed for rapid vascularization of the construct, surmounting limitations due to the diffusion limit present in solid implants. A summary of TE ovary models with in vivo evaluation is provided in Table 1. The greatest roadblock for all models is vascularisation. If the implanted graft cannot vascularize rapidly, follicles will die. All current TE ovary models have shown significant follicle mortality, most likely due to the nutrient diffusion barrier of non-vascularised grafts; however, several reports show restoration of endocrine function [42,45,47,49,52,53,56] and successful spontaneous pregnancy for grafts implanted in ovarian bursa [42,45,49,53], or IVF in the case of a graft under the kidney capsule [52]. 

While there are currently abundant studies on murine TE ovary models, there is a discernible lack of human studies with clinical translatability. Studies were carried out where human follicles were implanted in mice with functional ovaries [46,51]. The exposure to the mouse endocrine hormones meant that grafts could not be evaluated for endocrine function or fertility potential. Although further research is needed before clinical translation, especially for confronting the vascularisation problem, TE ovaries are promising for reversion of ovarian failure, especially in the case of cancer patients. Pre-vascularisation could be considered as a solution, such as presented by Jakubowska et al. in vaginal tissues [57], where endothelial cells were seeded within the scaffold to promote a rapid inosculation of the graft and host vessels. A functional TE ovary could eventually allow genetic motherhood via spontaneous pregnancy in cancer patients with failed ovaries, or IVF for patients without an intact ovarian bursa or uterus. Restoration of ovarian endocrine function would permit ovarian failure patients to avoid hormone therapy with potential long term cardiovascular effects.

### 2.3. Uterus

#### 2.3.1. Anatomy

The uterus is a pear-shaped organ about 7.6 cm long, 4.5 cm wide, and 3 cm thick [58]. The uterus is the main hormone-responsive secondary sex organ of the gynecological tract. Located in the pelvic area behind the bladder and in front of the rectum, the uterus is formed during embryogenesis by the fusion of the two Müllerian ducts to become a single, hollow organ. The lower part of the uterus, the cervix, opens to the vagina, while the upper part, the fundus, is connected to both fallopian tubes, leading to each ovary. The uterus is responsible for embryo nourishment and support as well as hormone regulation and menstruation. The body of the uterus, the corpus, has walls made up of three layers: the endometrium, the myometrium, and the serosa [59]. The endometrium is the inner lining and varies significantly in thickness and structure during menstrual cycles. The endometrium is responsible for embryo attachment and hormone responsiveness of the uterus. Comprised principally of epithelial, stromal, and gland cells, as well as a significant vascularisation (Figure 3A), the endometrium’s histology varies greatly in response to sex hormones. Estradiol and progesterone, secreted by the ovaries, stimulate proliferation, vilification of epithelial cells, and hormonal changes in preparation for potential embryo attachment. Post menstrual cycle, necrosis due to estradiol and progesterone absence leads to tissue and blood expulsion [60]. The myometrium is the middle and thickest layer of the uterus and is essentially comprised of smooth muscle cells. The myometrium is responsible for uterine structure and contractility. The serosa is the external lining of the uterus consisting of a thin layer of connective tissue and a superficial layer of mesothelium.

#### 2.3.2. Pathologies

Aside from primarily uterus-related pathologies, female infertility has been related to various comorbidities such as mental health disorders, gynecological malignancy and breast cancer, and cardiovascular disease [61], weighing on global healthcare systems. Women having undergone assisted fertility treatment show significantly less risk of such comorbidities, proving such treatments to not be of secondary importance. Congenital uterine malformation prevalence has been estimated at 6.7% of females [62]. These malformations are typically related to the incomplete fusion of the Müllerian ducts during embryogenesis and are known as Müllerian malformations. Severity of these malformations can range from absolute uterine-factor infertility as is the case with vaginal agenesis in patients with Mayer–Rokitansky–Küster–Hauser (MRKH) syndrome, to not requiring intervention for patients presenting an arcuate uterus, where only the final reabsorption of the uterine sinus has failed [63]. Risk factors during pregnancy are elevated for these patients. Prevalence of malformations has been shown to trend along with the organogenesis timeline, where the more severe the malformation, the less common it is. A prevalence ratio of 17:7:1 for arcuate, septate, and bicornuate malformations, respectively, was found after review of the literature [62]. 

Uterine and cervical cancer is the fourth most prevalent cancer in women globally, with an incidence of 13.3 out of 100,000 women in 2020 [31]. Of reproductive-aged women (15–49 years old), the prevalence increases to the second most common cancer with an incidence of 12.9 per 100,000 women [31]. 

Other common pathologies of the uterus can cause heavy or abnormal bleeding, and pain during intercourse and during defecation. Fibroid development affects 4.9–9.8% of reproductive-aged women [64]. Pelvic organ prolapse (POP) prevalence increases with age and parity. Defined by symptoms, 3–6% of women will be diagnosed with POP in their lifetime. This number increases to up to 50% for examination-based diagnosis [65]. Endometriosis is the abnormal growth of endometrial tissue and may also cause infertility. An incidence of 163 per 100,000 women globally was reported in 2017 [66]. Heavy bleeding and pelvic pain may also occur with no apparent reason, requiring medical intervention.

#### 2.3.3. Current Treatment

Müllerian malformations do not always require intervention. Women with an arcuate uterus rarely require resection of abnormal tissue. Conception is often not impaired; however, women with uterine malformations may experience recurring miscarriages. Surgical intervention is called for in cases of impeded menstrual blood or recurring miscarriage [63]. In the case of MRKH patients, there is currently no treatment option to restore uterine fertility. 

Except for minimally invasive, early-stage cervical cancer, where fertility may be preserved, the gold standard in endometrial and cervical cancer treatment is radical hysterectomy [67,68]. As early detection methods are generally reserved to wealthy countries, endometrial and cervical cancers are often detected later in developing countries, eliminating the chance of fertility preservation.

Hysterectomy is one of the most common surgeries for women globally. In the United States, 600,000 operations are performed annually, and one in nine women are estimated to undergo a hysterectomy in their lifetime [69]. Indications for the procedure include fibroid development (30%), heavy bleeding (20%), genital prolapse (15%), endometriosis (20%), or chronic pelvic pain (10%) [70]. Although there has been a net decrease in hysterectomies in recent years, expanding alternative treatment options is key to avoid unnecessary complications related to hysterectomies. Especially for pre-menopausal women and women of childbearing age, the risk of ovarian failure, premature menopause, and subsequent comorbidities should be avoided.

For patients presenting absolute uterine-factor infertility who have functional ovaries, such as is the case for MRKH patients, and for patients having undergone a partial or total hysterectomy, the only assisted fertility treatment option available is in vitro fertilisation followed by surrogacy to carry the foetus to term. This treatment comes with a plethora of ethical and legal barriers, where availability depends on the surrogacy laws and socioeconomic barriers of each country [71]. Surrogacy can be quite costly to individuals and carries cultural and religious barriers. 

#### 2.3.4. Alternative Solutions

Although the uterus is not a vital organ, women without a functional uterus experience an inferior quality of life as well as various associated comorbidities [61]. The scientific community has understood this; thus, the continuous reduction in hysterectomy incidence [69]. Nevertheless, continuous efforts should be made to improve the prognosis of women facing absolute uterine-factor infertility.

##### Uterus Transplantation

In 2013, the first live birth after a uterus transplant (UTx) was recorded [72]. This feat marked the first proof-of-concept for UTx as a viable fertility treatment for women with absolute uterine-factor infertility. To date, UTx has proved successful with both live and dead donors. As of 2021, at least two clinics perform UTx outside of clinical trials and at least 31 live births have succeeded post-transplantation [73], not taking into account unreported cases. It was shown that that 97.5% of women with absolute uterine-factor infertility see UTx as the first option for fertility treatment, before surrogacy [4]. Of those having undergone UTx, a 5-year study showed that women gained a sense of normalcy even just by having regular menstrual blood [74]. However promising this is as a possible fertility treatment, a critical perspective must be held for such novel techniques. The UTx technique requires significant optimization before wide-scale implementation may take place. The current success rate of UTx as a fertility treatment is currently insufficient. Successful live birth rates after UTx + IVF have been reported from 42–79% [73]. Reasons for failure are not yet fully understood, as success rates are based on technically successful UTx. It should be taken into account that around 28.6% of grafts require emergency hysterectomy, mainly due to thrombosis in the first 15 days postoperatively [75]. A disadvantage of the UTx fertility treatment is the constant threat of graft rejection. Women must take antirejection drugs throughout, risking potential side-effects for both the mother and the foetus (Figure 3B). This is why hysterectomy is prescribed after one to two live births, once the mother has finished reproducing.

##### Tissue Engineering

Throughout the 2000s, tissue engineering models have been developed with the goal of creating an artificial uterus, both as study models and in the scope of regenerative medicine [76,77]. To date, all models rely on scaffolds to guide cell architecture. For in vitro studies, most models have relied on collagen, often used together with Matrigel. Nearly all models aim only at endometrium reconstruction, ignoring the complexity of the three-layered tissue [77]. Few studies have been carried out in vivo. Table 2 summarises in vivo studies that observed an effect on fertility preservation. Most models implement decellularized uterine tissue as a scaffold (Figure 3C) [78,79,80,81]. Detergents are used to remove cells from the extracellular matrix, leaving a histologically intact scaffold with in vivo—like mechanical properties. Currently, no whole uterus model has been attempted. The majority of trials implanted small scaffold patches, acellular [78,81] or cell-seeded [79,80,82]. In all trials, late-stage pregnancy was achieved. The functionality of such TE uterine patches should be questioned, as for most studies it is unclear if embryo implantation occurred on patch site. As most embryo implantations were carried out at one month after grafting, it is doubtful that these patches, especially acellular patches, could effectively implant and nourish an embryo. Campbell et al. [83] interestingly attempted a completely autologous model, where a boiled blood clot was implanted in the peritoneal cavity as an “in vivo bioreactor”. The rat’s own myofibroblasts were allowed to colonize the scaffold, never passing in vitro before autografting the tissue to replace an excised uterine horn tip. Pregnancy attempts at 4-, 6-, and 12-weeks post grafting showed the tissue to be functional after 12 weeks, allowing for embryo attachment in implanted tissue. The functionalization of this graft, similar to an acellular scaffold patch, is entirely dependent on migration of endometrial and myometrial cells to colonize the scaffold. The closest to a whole uterus model attempted was a subtotal TE uterus implant, where a small strip of the uterine horn was left intact to attach the substitute to in a rabbit model [84]. A PGA/PLGA synthetic scaffold was seeded with primary endometrial and myometrial cells, and successfully restored fertility with term pregnancies achieved.

An in vitro model of interest achieved a full-thickness uterine wall capable of embryo implantation and maturation to blastocysts by combining scaffold use with self-assembly. Lü et al. [85] formed a myometrial smooth muscle layer in a collagen/ Matrigel scaffold, then seeded the surface with a fibroblast/epithelial cell mixture in a collagen/Matrigel scaffold. The result was improved histoarchitecture and embryo implantation compared to subsequent seeding of three separate tissue layers. Such a model would be interesting to assess in vivo in order to observe integration with native tissue and avoid reliance on migration of native cells. In hysterectomy patients or in MRKH patients, no uterus may be available to graft to. 

Two uterine models used bone marrow mesenchymal stem cells (BM-MSC) [79,82] for uterine tissue engineering. This choice of cell may be interesting for the immunomodulatory properties of the cells, although it is unclear if the uterine patches were able to support embryo attachment, possible due to uncontrolled differentiation [25]. It has been shown that MSC may be isolated in sufficient quantity from menstrual blood [86,87]. This may prove interesting for uterine tissue engineering due to uterine origin of cells, and avoid the painful BM-MSC collection procedure.

Current TE techniques for reconstruction of uterine tissue rely mostly on exogenous materials as scaffolds. Any exogenous material carries some risk of immune response and foreign body reaction. For decellularized organs especially, attention should be made to ensure complete decellularization, as any contamination risks implant rejection. In the case of uterine TE, any decellularized organ used would be allogenic of origin, carrying inherent risk of rejection. A TE uterus could eventually present a solution for absolute uterine-factor infertility that circumvents the need for a donor uterus and the numerous risks associated or the use of a surrogate mother. An autologous TE uterus could eliminate risk of rejection and the complicated ethical problems related to these methods. Further studies on larger animals, specifically non-human primates, are needed to bring us closer to any clinical trial as murine and rabbit models present significant physiological differences to humans. 

### 2.4. Vagina

#### 2.4.1. Anatomy

The vagina is the elastic muscular tubular organ connecting the cervix to the vulva with a depth of 7–15 cm. The vagina is mainly responsible for coitus, menstrual blood evacuation, and natural childbirth, but also plays a key role in microbiome transfer to newborns, which is impeded during cesarian section [88]. The vagina develops near the end of the first trimester from the fused Müllerian ducts and the paired sinovaginal bulbs to form its lumen [89]. The vaginal wall consists of three layers: the inner surface mucosa, the muscularis propria, and the adventitia (Figure 4A). The stratified squamous vaginal epithelium lies on the lamina propria and thickens with intracytoplasmic glycogen in the superficial layers under the influence of estrogen [89], thus thickening when hormones peaks during menstruation. The postnatal epithelium atrophies shortly after birth in the lack of placental estrogen and remains as such until puberty. The post-menopausal vaginal epithelium atrophies due to dwindling estrogen levels; keratinisation of the surface of the vaginal epithelium takes a phenotype more similar to that of the epidermis [90]. The muscular layer of the vagina extends from the uterus, throughout the vagina. Outer fibers are aligned longitudinally with the uterus, while the inner fibers are aligned in a spiral formation for a robust structure [89]. The stratified squamous vaginal epithelium lies on the lamina propria and thickens with intracytoplasmic glycogen in the superficial layers under the influence of estrogen [82], thus thickening when hormones peak during menstruation. The postnatal epithelium atrophies shortly after birth in the lack of placental estrogen and remains as such until puberty. The post-menopausal vaginal epithelium atrophies due to dwindling estrogen levels; keratinisation of the surface of the vaginal epithelium takes a phenotype more similar to that of the epidermis [83] Lactobacilli colonization of the vagina creates an acidic microenvironment through the production of lactic acid, protecting from potentially pathogenic bacteria and viruses, such as Human Immunodeficiency Virus (HIV) [91,92].

#### 2.4.2. Pathologies

Various defects and pathologies may affect the vagina, either congenital or acquired. Failed or incorrect midline fusing of the pelvic structures (bladder, genitals, colon) may cause bladder and cloacal exstrophy amongst other malformations [93]. Intersex disorders such as adrenal hyperplasia and cloacal abnormalities can be significant defects, often requiring external tissue sources for surgical reconstruction [94]. 

As previously mentioned, MRKH is the agenesis of Müllerian structures. This not only causes agenesis of the uterus, but of the superior 2/3 of the vagina as well [95,96]. As the sinovaginal bulbs have no Müllerian structure to fuse to, improper canalization occurs, leaving patients with shallow vaginas often no deeper than 3.5 cm, sometimes with no depth at all [97]. Outer vaginal structures appear as normal, meaning that diagnosis usually does not occur until puberty when patients observe the lack of menstrual blood (primary amenorrhea). Associated symptoms such as renal complications are often present [95]. 

Various cancers proximal to the vagina may be cause for partial or total vaginal resection such as cervical, uterine, ovarian, rectal, bladder, or vaginal cancers. Pelvic radiotherapy for such cancers is also known to cause vaginal stenosis, the narrowing and shortening of the vagina due to fibrosis in up to 88% of patients [98]. Vaginal stricture may also be caused by vaginal atrophy, hypoestrogenic states, inflammatory and autoimmune diseases, and chemical vaginitis [99]. Vaginal birth may also perturb normal vaginal structure, temporarily or long-term. Most women present damaged supporting tissues postpartum [100]. Stress urinary incontinence (SUI) is often a result of such structural damage with a prevalence at 65% of women, increasing with parity and age [101]. Significant psychosexual distress is related to vaginal defects, such as MRKH, and has been shown to improve with treatment [102,103]. 

#### 2.4.3. Current Treatments

Vaginal malformations may require either surgical or non-surgical corrections depending on the severity of the malformation. Over the past 100 years, various techniques have been suggested for vaginal construction. Non-surgical techniques are related with the least risk and rely on dilation of the existing dimple to create a neo-vagina. Currently, the first treatment approach is Frank’s technique where dilators of gradually increasing size are inserted for 10–30 min, 1–3 times a day by the patient (Figure 4B top) [104]. An adaptation of this method is Ingram’s technique which employs a bicycle seat to hold the dilators so the patient’s hands may be freed during treatment, using their own weight to exert pressure [105]. An alternative to self-dilation is d’Alberton’s method, dilation by coitus, which can give satisfactory results but requires frequent coital activity and is not suitable for everyone [106]. These dilation techniques are unsuitable for patients with a vaginal dimple less than 3–4 cm, which is often the case with MRKH. Although results are often satisfactory, the limitations are discomfort, need for perseverance and a significant amount of time, and the need to wait for sexual maturity [107]. 

Surgery may provide an alternative option for vaginal reconstruction for patients with insufficient dimple depth or for those having attempted dilation and failed. The Vecchietti technique [108] is a minimally invasive method based on vaginal dilation where an acrylic olive in the vaginal dimple is connected to the abdominal wall laparoscopically using a tensile device. The tension is adjusted frequently to form a neovagina in a matter of days/weeks. The advantage of this method, similar to non-surgical dilation, is that the vaginal mucosa is preserved as entire neovagina, as well as conservation of vaginal flora. Limitations include discomfort and the relatively higher risk of long-term contraction, prolapse, and urological lesions.

As early as 1898, Abbè’s vaginoplasty was introduced, using an autologous skin graft from the inner thighs for reconstruction [109]. The McIndoe method builds on this by wrapping the skin graft around a stent and is currently the most widespread surgical alternative to vaginal dilation. Various other methods use autologous tissues such as oral mucosa, bowel sections, or vulvar flaps as reconstructive tissue (Figure 4B bottom) [107]. Avoiding the use of dilators lends these surgeries to pediatric use. Problems associated with use of autologous tissue grafts lies in the innate differences in tissue phenotypes. McIndoe’s method often leads to vaginal dryness and hair growth, leading to painful intercourse. Bowel sections often excrete excessive mucus, causing odor problems. 

#### 2.4.4. Tissue Engineering

Several TE methods have been introduced for reconstruction of vaginal tissue with autologous cells, some trials having proceeded all the way to a clinical setting [110]. Table 3 summarizes in vivo TE vagina models to date. The first attempt at implanting a vaginal replacement used decellularized vaginal or bladder tissue without recellularization in vitro prior to implantation [111]. Although re-epithelialization occurred, the graft collapsed shortly after implantation. All subsequent TE vaginal substitutes have been cell-seeded in vitro before implantation. De Filippo has reported two attempts of in vivo implantation, one subcutaneous murine model [112], and one in situ rabbit study [113]. Native histoarchitecture was recreated by seeding vaginal epithelial and smooth muscle cells on opposite sides of a PGA/ PLGA scaffold, then maturing the construct in a bioreactor before implantation. Up to 6 months after implantation, the TE vagina was fully functional with native-like histology. The synthetic scaffold had completely degraded. Raya–Rivera published a clinical trial carried out by the same team as De Filippo, implanting their constructs in four women [114]. Follow-ups were carried out up to 8 years post implantation and appeared promising as a functional TE vaginal substitute, with native-like histology. Despite seemingly promising results, the trial was not continued with a larger cohort, and no other subsequent clinical trials have taken place to our knowledge. 

Scaffold use in vaginal TE is promising, with decellularized tissues and synthetic hydrogels having been proven in vivo models as potentially viable reconstruction methods. Most of these models were constructed over a decade ago, with no progress made since the clinical trial which only had four participants. More recent vaginal TE attempts were carried out by Orabi [115] and Jakubowska [57]. These models used scaffold-free cell sheets, taking advantage of the capacity of fibroblasts to produce their own ECM in the presence of ascorbic acid, a technique known as self-assembly, pioneered by Dr. François

Auger [116]. Human vaginal fibroblasts (hVF) and human vaginal epithelial cells (hVEC) were extracted from biopsies. The hVF were cultivated in the presence of ascorbate for 4 weeks in 6-well plates. Once the stroma was strong enough to be manipulated, the hVEC were seeded on top of the constructs and cultivated until the upper surface was fully confluent (1 week). The reconstructed tissue was then elevated to the air/liquid interface for 3 additional weeks to obtain a mature epithelium (Figure 4C). This tissue possessed many histological and molecular characteristics of the native vaginal mucosa, notably the presence of a layer of glycogen-storing cells [115]. The engineered tissue was also hormone responsive, increasing its thickness in the presence of estradiol, and has served as model for HIV infection [117]. Jakubowska, continued the work of Orabi by pre-vascularizing the vaginal substitutes, and prolonging in vivo observation to 3 weeks, proving the presence of several vaginal mucosa-specific markers, as well as natural lubrification. The advantage of this technique is that risk of rejection is significantly reduced, as the entire implant is autologous with no exogenous material. Decellularized tissue scaffolds especially present a risk of rejection if tissue is not sufficiently cleared of immunocompetent material.

Interestingly, self-assembly tissue engineering could prove promising as an alternative to the current void in adequate treatment for pelvic organ prolapse (POP) and for stress urinary incontinence (SUI). The gold standard treatment for both of these disorders relies on surgical implantation of support materials, most often polypropylene mesh. A warning was issued by the FDA for these synthetic mesh implants, due to the elevated risk of surrounding tissue erosion. This led to severe restrictions and banning of the material in several countries. Adequate replacement materials are under research [23,118]. Self-assembly could provide anatomically relevant structural support tissue with little risk of immune response [119]. 

The advantage of TE vaginal substitutes is that tissue-specific implants can be created, conserving the properties of a native vagina such as mucus production of microbiome. Disadvantages of current techniques could be avoided such as long-term trauma from dilation or issues related to non-vaginal autografts such as lack of mucus from skin grafts or excess odorous mucus from grafting colon segments. As vaginal construction/reconstruction is not generally an urgent treatment, the time necessary for biopsy/cell culture and TE vaginal construction is of minimal importance compared to potential benefits. Vaginal dilation is a lengthy process and is not necessarily faster than vaginal TE. Tissue engineering could eventually serve as a solution for persons with intersex disorders or for trans women as well. 

## 3. Conclusions

A significant proportion of the population suffers from infertility and gynecological organ dysfunction due to congenital diseases and acquired factors related to modern lifestyles. Cancer treatments in particular, have a high risk of inducing organ failure. Novel alternatives to current treatment methods have been in development since the 1990s in attempts to restore native function to affected women. Although murine models are quite developed and prove promising for ovarian, uterine, and vaginal reparation/ replacement, almost none have been brought to a clinical setting. Further studies in nonhuman primates would provide pertinent results, allowing TE organ substitutes to be brought to a clinical setting. Avoiding allogenic materials with risk of rejection, and surrogate mothers, which poses significant ethical questions, could improve fertility medicine, an increasingly important domain as reproduction numbers dwindle in the developed world. 

## Figures and Tables

**Figure 1 ijms-23-12319-f001:**
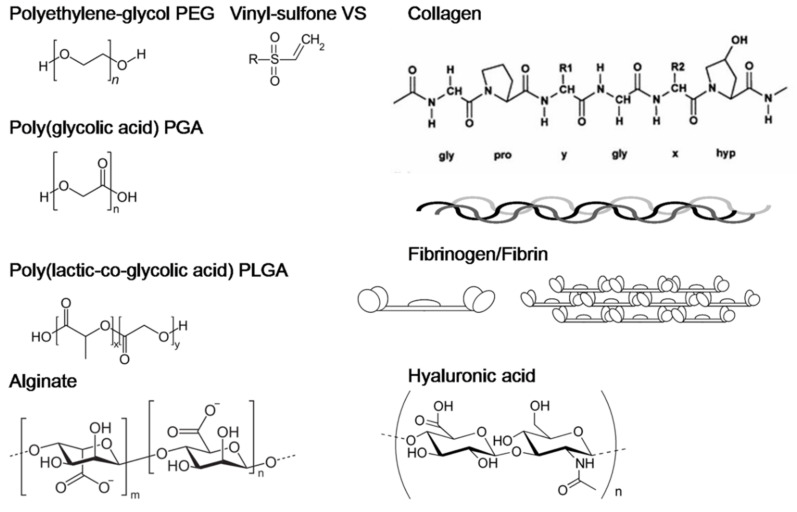
Schematic representation of polymers used to create scaffolds used in vivo for female genital tissue engineering. Vinyl-sulfone can combine in several occurrences with PEG. Polyglycolic acid and PLGA can be combined to support growth of cells during their biodegradation. These synthetic biopolymers rely on hydrophilic interaction for cell adhesion and may be crosslinked to various degrees to control substrate stiffness. Alginate can be combined with other material to serve as a support while being relatively inert in terms of biological function. Collagen and its degraded form gelatin are widely used in tissue engineering due to their being the main protein component of the ECM. Collagen is composed of three polypeptide chains to form a triple-helical configuration consisting of proline (pro), lysine (lys) or their hydoxylated forms such as hydroxyproline (hyp). Fibrin is derived from fibrinogen present in blood. Fibrinogen consists of nodules situated at the extremity of the molecule, around a central node. Enzymatic activation of the nodules by thrombin induces the polymerisation of the fibrin to form a blood clot.

**Figure 2 ijms-23-12319-f002:**
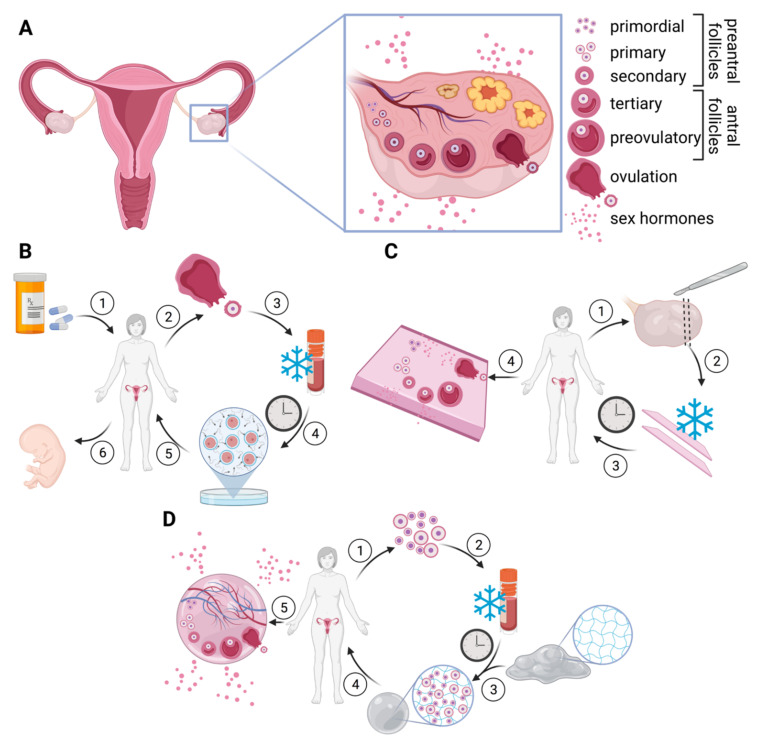
Prospective treatments for ovary failure: (**A**) Histoanatomy of the ovary. Ovulation occurs after follicle maturation triggered by the menstrual cycle. (**B**) In vitro fertilization (IVF) is the current fertility preservation method for women undergoing cancer treatment. Oocytes are harvested after drug-induced ovulation and kept in cryostorage until ready for IVF treatment. (**C**) Ovarian tissue cryopreservation is increasing in prevalence. Thin slices of ovarian tissue are harvested and cryopreserved until after cancer treatment. The tissue is implanted orthotopically for ovary functionality restoration or heterotopically for endocrine restoration and possibility of IVF. (**D**) Tissue engineering could allow ovary replacement. Preantral follicles are harvested and cryopreserved ready for ovary replacement. Hydrogels are generally used as a scaffold to create an artificial organ that can be implanted in situ. Fertility has been restored in murine models along with endocrine function. Numbers indicate procedure order. Figure created with BioRender.com (accessed on 1 June 2022).

**Figure 3 ijms-23-12319-f003:**
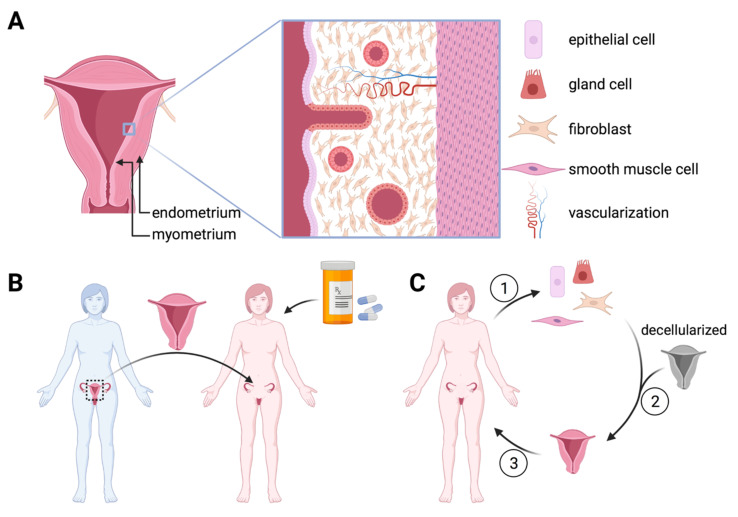
Prospective treatments for uterine factor infertility: (**A**) Histoanatomy of the uterus. The highly vascularized endometrium is covered by a columnar epithelium atop a stroma containing many glands and lies above the muscular myometrium. (**B**) Uterine transplant (UTx) is emerging as a potential treatment for uterine factor infertility. A live or cadaveric donor (blue) uterus is transplanted to the patient (pink) who takes antirejection drugs until childbearing is finished. (**C**) Tissue engineering could restore uterine function. Murine models have restored uterine function by seeding a decellularized uterus with autologous uterine cells. Numbers indicate procedure order Figure created with BioRender.com.

**Figure 4 ijms-23-12319-f004:**
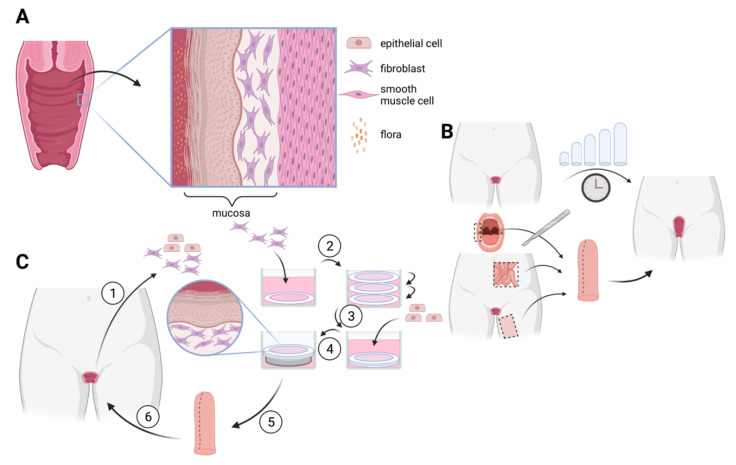
Techniques for neovagina formation: (**A**) Histoanatomy of the vagina. A glycogenated stratified squamous epithelium lies atop a stroma forming the vaginal mucosa which is above the muscularis of uterine origin. (**B**) Vaginal dilation is the first line of treatment for MRKH patients (**top**). Surgical options are available, relying of heterotopic tissue substitutes. (**C**) Tissue engineering could allow for autologous vaginal mucosa grafting in MRKH patients. The “self-assembly” method is scaffold-free as fibroblasts secrete ECM components in the presence of ascorbic acid. Fibroblast-ECM sheets are fused and seeded with epithelial cells for histologically similar autologous tissue in vitro that has shown promising results in murine models. Numbers indicate procedure order. Figure created with BioRender.com.

**Table 1 ijms-23-12319-t001:** Tissue engineering ovary models in vivo.

Reference	Scaffold	Host	Follicles	Sterilization	Graft Location	Endocrine Function	IVF	Spontaneous Pregnancy	Results
Gosden 1990 [42]	plasma clot	mouse	mouse	x-ray or ovariectomy	ovarian bursa	yes	n/a	yes	normal follicular development
Telfer et al. 1990 [52]	collagen	mouse	mouse	ovariectomy	kidney capsule	yes	yes	n/a	normal follicular development
Carroll, Gosden 1993 [45]	plasma clot	mouse	mouse	ovariectomy	ovarian bursa	yes	n/a	yes	normal follicular development
Dolmans et al. 2008 [46]	plasma clot	mouse	human	n/a	ovarian bursa	n/a	n/a	n/a	antral follicle development
Vanacker et al. 2012 [55]	alginate/ matrigel	mouse	mouse	ovariectomy	peritoneum	n/a	n/a	n/a	graft–host integration
Vanacker et al. 2014 [54]	alginate	mouse	mouse	ovariectomy	peritoneum	n/a	n/a	n/a	antral follicle development
Smith et al. 2014 [47]	fibrin	mouse	mouse	ovariectomy	ovarian bursa	yes	n/a	n/a	antral follicle development
Rajabzadeh et al. 2015 [48]	fibrin	mouse	mouse	ovariectomy	subcutaneous neck	n/a	n/a	n/a	antral follicle development
Kniazeva et al. 2015 [49]	fibrin fibrin/collagen fibrin/alginate	mouse	mouse	ovariectomy	ovarian bursa	yes	n/a	yes	sustainable ovary graft
Paulini et al. 2016 [51]	fibrin fibrin/ HA	mouse	human	n/a	peritoneum	n/a	n/a	n/a	secondary follicle development
Kim et al. 2016 [56]	PEG-VS	mouse	mouse	ovariectomy	ovarian bursa	yes	n/a	n/a	antral follicle development
Laronda et al. 2017 [53]	3D printed gelatin	mouse	mouse	ovariectomy	ovarian bursa	yes	n/a	yes	antral follicle development

**Table 2 ijms-23-12319-t002:** Summary of tissue-engineering uterus models in vivo.

Reference.	Tissue	Host	Scaffold	Cells	Time In Vitro	Time In Vivo	Time of Conception	Gestation Achieved
Santoso et el. 2014 [78]	full thickness uterine patch	rat	decellularized rat uterus	acellular	n/a	≤51 days	30 days	late-stage
Miyazaki & Maruyama 2014 [79]	full thickness uterine patch	rat	decellularized rat uterus	rat neonatal, adult uterine cells and rat BM-MSCs	≤10 days	≤90 days	28 days	late-stage
Ding et al. 2014 [82]	full thickness uterine patch	rat	collagen	rat BM-MSCs	72 h	105–109 days	90 days	late-stage
Hellström et al. 2016 [80]	full thickness uterine patch	rat	decellularized rat uterus	rat uterus primary cells and rat mscs	3 days	9 weeks	6 weeks	late-stage
Hiraoka et al. 2016 [81]	full thickness uterine patch	mouse	decellularized mouse uterus	acellular	n/a	≤7 weeks	4 weeks	full term
Campbell et al. 2008 [83]	full thickness uterine horn tip	rat	boiled blood clots	uncontrolled host cellularization	in vivo 2 weeks	4–12 weeks	4,6,12 weeks	late-stage
Magalhaes et al. 2020 [84]	subtotal uterus	rabbit	PGA/PLGA	rabbit primary endometrial and myometrial cells	--	≤7 months	6 months	full term

**Table 3 ijms-23-12319-t003:** Summary of tissue-engineering vagina models in vivo.

Reference	Host	Scaffold	Cells	Time In Vitro	Time In Vivo	Graft Site	Results
Wefer et al. 2002 [111]	Rat	Decellularized vagina or bladder	acellular	N/A	2–12 weeks	In situ	Scaffold re-epithelialization, graft collapse
Raya–Rivera et al. 2014 [114]	Human	Decellularized porcine intestinal submucosa	Human vulvar epithelial and smooth muscle cells	7 days	≤8 years	In situ	Functional organ with native histology, Sexual satisfaction
De Filippo et al. 2003 [112]	Mouse	PGA/PLGA	Rabbit Vaginal epithelial and smooth muscle cells	24–48 h	1–6 weeks	subcutaneous	Histology restauration, vascularization and innervation
De Filippo et al. 2008 [113]	Rabbit	PGA/PLGA	Rabbit Vaginal epithelial and smooth muscle cells	10 days	1–6 months	In situ	Good histology and tensile strength, innervation
Orabi et al. 2017 [115]	Mouse	Scaffold-free stromal cell ECM sheets	Human vaginal epithelial and stromal cells	7 weeks	1–2 weeks	subcutaneous	Native-like histology, expression of vaginal markers
Jakubowska et al. 2020 [57]	Mouse	Scaffold-free stromal cell ECM sheets	Human vaginal epithelial and stromal cells and HUVEC	8 weeks	≤21 days	subcutaneous	Native like histology, integration with host tissue, improved vascularization with HUVEC

## Data Availability

Not applicable.

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
