# Peer review of "Tissue Engineering in Gynecology"

_ijms, 2022, doi:10.3390/ijms232012319_

Round 1

Reviewer 1 Report

The article is a review on the current state-of-the-art in TE in gynaecology. Leading into the problem with an overview over the most prevalent reasons for and effects of female infertility, it then segues into the practical aspect and first introducing the biomaterials used before going into detail for the different organs. For each of the organs the authors first explain the anatomy and pathologies and continue with current and then alternative treatments, among them TE. This overall structure means the reader can follow the line of argumentation very well.

I do feel a little less emphasis on the first three parts and some more thorough descriptions on important studies on the TE would be beneficial for a review titled “TE in gynaecology”. It would be especially helpful to have some images from successful TE implants, possibly incorporated into the overview figures you already have

General open questions / suggestions:

Introduction: You very much concentrate on infertility, but what about women mutilated by “circumcision”? (How) can TE help in restoring these women’s quality of live? Especially since you discuss the vagina in-depth later on anyway, it might be an idea to also include this in the introduction

Line 96 ff: You first explain alginate, then introduce the (dis)advantages of natural biopolymers before explaining collagen. As alginate is definitely also a natural polymer maybe switch the order?

Figure 1: Your figure description is not quite consistent, for some materials you explain how they can be combined, for others their structure. Could you make this more consistent (one or the other, or both for all)?

Line 167 f: “if efforts are not made to preserve fertility”. Could you expand on that sentence a bit, maybe give an example? Because one would always hope that “efforts are made”, and read with a cynical mind the sentence sounds like “infertility only comes about if you have an uncaring surgeon in such a case”

Line 397: You write “late term pregnancy was achieved”, as far as I know late-term pregnancy is defined as a pregnancy that lasts too long which can be quote hazardous and is not at all something to be desired, so why would this be an achievement? Or did you mean late-stage, as in “most of the normal time of a pregnancy has passed but the embryo did not develop fully”? If so it would be best to ads sth like “but could not be carried to term” or similar

Line 490: Please also include the actual figure 4 not only its description

Line 507ff: While a fun little tidbit, the internet does not seem to have anything on the Albertson’s method except for that rather short letter that you cite – do you have any other sources on that? Otherwise, I’d maybe not write that in a way that it sounds like a routine method, but rather like the theory it seems to be

Language / typos:

-       Line 70: OtherS

-       Line 118: “extermity”? I assume you mean “extremity” and shouldn’t it also be plural? Possibly reconsider replacing by sth like “at both ends” as I only know extremities used in an anatomical sense (I’m always open to learning new things though…)

-       Line 139: remove the “se”

-       Figure 2: “queued by”. Correct me if I’m wrong, but I only know this specific wording from IT, in the case of ovaries I personally would use the simple “followed by”

-       Line228: Sign should be plural I think

-       Line 236: I would suggest “malleability” instead of “manipulability”

-       Line 249: restoration

-       Line 254 f: Please rephrase, the studies were not implanted

-       Line 285: remove the full stop in the middle of the sentence

-       Line 373: “is as a treatment”

-       Line 494: method

-       Line 456: peak

-       Line 533: reconstruction of

Reviewer 2 Report

The presented paper describes the pathologies and current treatments of gynecological organs (ovaries, uterus, and vagina). Despite the overall good impression of the submitted paper, the following remarks should be corrected.

1. The paper does not disclose the use of anti-inflammatory properties of MSCs-based therapy.

2. The authors should note not only the issue of genital bioengineering, but pelvic organ prolapse (POP) and female pelvic floor reconstruction. The manuscript should disclose the issue of the use of meshes seeded with cells for pelvic floor reconstruction and augmentation:

2.1. Gargett, C. E., Gurung, S., Darzi, S., Werkmeister, J. A., & Mukherjee, S. (2019). Tissue engineering approaches for treating pelvic organ prolapse using a novel source of stem/stromal cells and new materials. Current Opinion in Urology, 29(4), 450-457.

2.2. Wu, X., Jia, Y., Sun, X., & Wang, J. (2020). Tissue engineering in female pelvic floor reconstruction. Engineering in Life Sciences, 20(7), 275-286.

3. Another lacking topic is the applications of mesenchymal cells derived from menstrual blood for genital bioengineering:

3.1. Kovina, M. V., Krasheninnikov, M. E., Dyuzheva, T. G., Danilevsky, M. I., Klabukov, I. D., Balyasin, M. V., ... & Lyundup, A. V. (2018). Human endometrial stem cells: High-yield isolation and characterization. Cytotherapy, 20(3), 361-374.

3.2. Ulrich, D., Edwards, S. L., Su, K., Tan, K. S., White, J. F., Ramshaw, J. A., ... & Gargett, C. E. (2014). Human endometrial mesenchymal stem cells modulate the tissue response and mechanical behavior of polyamide mesh implants for pelvic organ prolapse repair. Tissue Engineering Part A, 20(3-4), 785-798.

4. In line #445 the authors refer to figure 4A. But this figure is missing from the manuscript.

5. Complications and adverse events of bioengineered tissues. Moreover, cell therapy does not always lead to completely tissue restoration. I recommend that a separate subchapter be devoted to this topic:

5.1. Drela, K., Stanaszek, L., Nowakowski, A., Kuczynska, Z., & Lukomska, B. (2019). Experimental strategies of mesenchymal stem cell propagation: adverse events and potential risk of functional changes. Stem cells international, 2019.

5.2. Maksimova, N. V., Michenko, A. V., Krasilnikova, O. A., Klabukov, I. D., Gadaev, I. Y., Krasheninnikov, M. E., ... & Lyundup, A. V. (2022). Mesenchymal stromal cell therapy alone does not lead to complete restoration of skin parameters in diabetic foot patients within a 3-year follow-up period. BioImpacts, 12(1), 51-55.
